# De Novo Large Deletions in the *PHEX* Gene Caused X-Linked Hypophosphataemic Rickets in Two Italian Female Infants Successfully Treated with Burosumab

**DOI:** 10.3390/diagnostics13152552

**Published:** 2023-07-31

**Authors:** Carmine Pecoraro, Tiziana Fioretti, Assunta Perruno, Antonella Klain, Daniela Cioffi, Adelaide Ambrosio, Diego Passaro, Luigi Annicchiarico Petruzzelli, Carmela Di Domenico, Domenico de Girolamo, Sabrina Vallone, Fabio Cattaneo, Rosario Ammendola, Gabriella Esposito

**Affiliations:** 1Paediatric Nephrology, Dialysis and Renal Transplantation Unit, Santobono Pausilipon Children’s Hospital, 80129 Naples, Italy; l.a.petruzzelli@gmail.com; 2CEINGE—Advanced Biotechnologies Franco Salvatore, 80145 Naples, Italy; fioretti@ceinge.unina.it (T.F.); ambrosio@ceinge.unina.it (A.A.); passaro.diego@libero.it (D.P.); didomenico@ceinge.unina.it (C.D.D.); degirolamo@ceinge.unina.it (D.d.G.); 3Primary Care Pediatrician, ASL NA2 North, 80027 Naples, Italy; perruno@libero.it; 4Pediatric Endocrinology Unit, Santobono Pausilipon Children’s Hospital, 80129 Naples, Italy; a.klain@santobonopausilipon.it (A.K.); d.cioffi@santobonopausilipon.it (D.C.); 5Department of Molecular Medicine and Medical Biotechnologies, University of Naples Federico II, 80131 Naples, Italy; sabrina.vallone99@gmail.com (S.V.); fabio.cattaneo@unina.it (F.C.); rosario.ammendola@unina.it (R.A.)

**Keywords:** burosumab, FGF23, heterozygous deletion, hypophosphatemic rickets, multiple ligation probe amplification, next-generation sequencing, *PHEX* gene, therapy, X-linked dominant hypophosphataemia

## Abstract

Pathogenic variants in the *PHEX* gene cause rare and severe X-linked dominant hypophosphataemia (XLH), a form of heritable hypophosphatemic rickets (HR) characterized by renal phosphate wasting and elevated fibroblast growth factor 23 (FGF23) levels. Burosumab, the approved human monoclonal anti-FGF23 antibody, is the treatment of choice for XLH. The genetic and phenotypic heterogeneity of HR often delays XLH diagnoses, with critical effects on disease course and therapy. We herein report the clinical and genetic features of two Italian female infants with sporadic HR who successfully responded to burosumab. Their diagnoses were based on clinical and laboratory findings and physical examinations. Next-generation sequencing (NGS) of the genes associated with inherited HR and multiple ligation probe amplification (MLPA) analysis of the *PHEX* and *FGF23* genes were performed. While a conventional analysis of the NGS data did not reveal pathogenic or likely pathogenic small nucleotide variants (SNVs) in the known HR-related genes, a quantitative analysis identified two different heterozygous de novo large intragenic deletions in *PHEX*, and this was confirmed by MLPA. Our molecular data indicated that deletions in the *PHEX* gene can be the cause of a significant fraction of XLH; hence, their presence should be evaluated in SNV-negative female patients. Our patients successfully responded to burosumab, demonstrating the efficacy of this drug in the treatment of XLH. In conclusion, the execution of a phenotype-oriented genetic test, guided by known types of variants, including the rarest ones, was crucial to reach the definitive diagnoses and ensure our patients of long-term therapy administration.

## 1. Introduction

Despite knowledge advances in the pathophysiology of rare inherited diseases, many of them do not have an approved treatment. Indeed, only a small fraction of the orphan drugs investigated to date obtain regulatory approval each year, and they only do so after passing rigorous preclinical and clinical studies [1].

The introduction of burosumab, the human monoclonal antibody directed against fibroblast growth factor 23 (FGF23) [2,3], marked a critical breakthrough in the treatment of inherited hypophosphatemic rickets (HR), a group of rare, genetically heterogeneous phosphate wasting disorders (prevalence of 3.9 per 100,000 live births) that impair bone mineralization and severely impact the quality of life of affected patients.

In association with calcium, phosphorus is required to maintain bone integrity and mineralization. The maintenance of intracellular and extracellular phosphate levels depends on a complex set of processes occurring in the gut, skeleton, and kidneys, which are regulated by parathyroid hormone (PTH), 1,25-dihydroxy vitamin D3 (1,25(OH)2D3) or calcitriol, and FGF23, a phosphaturic hormone produced by bone. PTH stimulates renal phosphate excretion, calcitriol increases the absorption of phosphate by the intestine and bones, and FGF23 suppresses proximal tubular phosphate reabsorption and intestinal phosphate absorption, controls calcitriol degradation, and suppresses PTH production and secretion. Hypophosphatemia can arise from inadequate phosphorus intake, reduced intestinal absorption, the redistribution of tissue fluid phosphorus into cells, or excessive renal wasting [4].

FGF23 excess or hyperactivity causes the waste of urinary phosphate and low levels of 1,25(OH)2D3, thereby leading to HR with characteristic lower limb deformity, growth plate abnormalities, and progressive softening of the bones (osteomalacia), as well as a short stature, muscle and bone pain, weakness, fatigue, joint pain or stiffness, hearing difficulties, enthesopathy, osteoarthritis, and dental abscesses [5]. Increased FGF23 activity may depend on (i) an abnormal overexpression, as in the case of the rare tumor-induced osteomalacia; (ii) specific pathogenic variants in the *FGF23* gene leading to mutant proteins that are resistant to enzymatic processing; and (iii) impaired FGF23 degradation due to partial or full deficiency of the phosphate-regulating endopeptidase homolog X-linked *(PHEX)* gene product [4].

Conventional treatment for HR consists of multiple daily doses of vitamin D analogs and phosphate salts administration. However, in the long-term, this therapeutic regimen has a wide range of side effects, including hypercalcemia, hypercalciuria, nephrolithiasis, nephrocalcinosis, gastrointestinal disorders, impaired renal function, and potential chronic kidney disease. Moreover, it may improve radiographic rickets, but it does not normalize growth. In contrast, burosumab has performed well in different trials in children and adults with X-linked hypophosphataemia and in patients with tumor-induced osteomalacia since it improves serum phosphorus levels and mineralization and decreases rickets severity and pain scores [2,3,4].

Despite pathogenic sequence variants in at least 20 genes have been to date associated with hereditary HR, thereby resulting in autosomal dominant, autosomal recessive, and X-linked dominant and recessive conditions [5,6,7,8], only the inherited forms of HR that are related to FGF23 hyperactivity can be treated with burosumab. These latter conditions include the X-linked dominant hypophosphataemia, which is associated with a mutation of the *PHEX* gene (OMIM: *300550); the autosomal dominant HR, which occurs due to a mutation of the *FGF23* gene (OMIM: *605380); the autosomal recessive HR 1 and 2 conditions, which are caused by pathogenic variants in the dentin matrix acidic phosphoprotein 1 (*DMP1;* OMIM: *600980) and ectonucleotide pyrophosphatase/phosphodiesterase 1 (*ENPP1;* OMIM: *173335) genes, respectively; osteoglophonic dysplasia, which occurs due to pathogenic variants in the fibroblast growth factor receptor 1 gene (*FGFR1*, OMIM: *136350); Jansen-type metaphyseal chondrodysplasia, which is associated with a mutation of parathyroid hormone 1 receptor (*PTH1R*, OMIM: *168468); Raine syndrome, which is associated with the Golgi-associated secretory pathway kinase *FAM20C* gene (OMIM: *611061); and McCune–Albright syndrome, which is associated with the guanine nucleotide-binding protein G(s) subunit alpha (GNAS) complex gene (*GNAS1*, OMIM: *139320).

The most common form of heritable HR related to FGF23 excess, affecting approximately 1:20,000 individuals, is X-linked dominant hypophosphataemia, aka X-linked hypophosphataemic rickets (XLH; OMIM: #307800), which occurs due to loss-of-function variants in the *PHEX* gene on chromosome Xp22.1 [5,8]. Consequently, both hemizygous males and heterozygous females can be affected by this lifelong progressive and severe metabolic bone disease. *PHEX* mutations have been found in 87% of familial cases and in 72% of sporadic cases as a result of de novo mutational events [9,10].

Next-generation sequencing (NGS) methodologies looking for small nucleotide variants (SNVs) in the genes currently associated with hereditary HR have strongly reduced the timing of diagnosis. However, conventional NGS analyses can miss copy number variations (CNVs), mainly represented by large intragenic deletions, especially in heterozygotes [11].

Herein, we report the clinical, radiological, and laboratory findings from two Italian female children affected by sporadic HR who successfully responded to burosumab treatment and underwent molecular analysis to obtain definitive diagnosis for therapeutic purposes and genetic counseling.

## 2. Materials and Methods

The clinical diagnosis of HR associated with FGF23 excess was formulated according to recent recommendations that have suggested evaluating the presence of muscle pain, weakness, and/or fatigue; lower limb deformities, fractures/pseudo-fractures, tooth abscesses and/or excessive dental caries; bone pain, joint pain, and/or joint stiffness; short stature and gait abnormalities; family history; serum calcium (normal range 8.4–10.2 mg/dL) and phosphate (normal range 3.4–4.5 mg/dL), alkaline phosphatase (ALP; normal range 44–147 U/L), 25-hydroxy vitamin D (25(OH)_2_D) (normal range > 30 ng/mL), and PTH (normal range 14–65 pg/mL) and levels of intact FGF23 (normal range 58.63 to 63.71 pg/mL); and renal functionality assessed by creatinine levels, the estimated glomerular filtration rate (eGFR; normal range > 90 mL/min/1.73 m^2^), and ultrasound [2,3,12,13].

Written informed consent was obtained from the parents prior to participation in the study. Peripheral blood samples were obtained from the probands and parents. Genomic DNA was extracted from peripheral blood leukocytes collected in EDTA with a QIAa–mp DNA Mini Kit (QIAGEN Italia, Milan, Italy) or by using a Maxwell 16 instrument (Promega, Madison, WI, USA). DNA concentration, purity, and integrity were evaluated using a Nanodrop spectrophotometer (Thermo Fisher, Waltham, MA, USA) and a Tape Station analyzer (Agilent, Santa Clara, CA, USA). For the NGS analysis of genomic DNA, library preparation was carried out with the multi-gene panel (4800 genes) SureSelect Clinical Research Exome V2, according to the manufacturer’s instructions (Agilent Technologies, Santa Clara, CA, USA). The libraries that passed quality control were sequenced by the paired-end sequencing-by-synthesis method with the NextSeq 500 sequencing system using the High Output PE 300 Cycles flow-cell (Illumina, San Diego, CA, USA). The paired-end reads of 150 and 100 bp of the lengths were generated in accordance with the supplied protocol. Genomic target regions were sequenced at a high-depth coverage (20x minimum, 150x average). FASTQ files were uploaded to the Alissa Clinical Informatic Platform (Agilent Technologies) to call, annotate, filter, and prioritize the variants, and a bioinformatics pipeline that incorporates community standards and custom algorithms was used to analyze the NGS reads and identify single nucleotide variants (SNVs) and small and large insertions/deletions (indels).

To assess variant pathogenicity according to the American College of Medical Genetics and Genomics and the Association for Molecular Pathology (ACMG) guidelines, we retrieved the annotations reported in the VarSome platform (https://varsome.com/ accessed on 30 June 2023) and/or ClinVar (https://www.ncbi.nlm.nih.gov/clinvar/ accessed on 30 June 2023). In agreement with the ACMG guidelines, variants can be classified as pathogenic (P), likely pathogenic (LP), variant(s) of uncertain significance (VUS), likely benign (LB), or benign (B). Only P/LP variants have to be considered to confirm a molecular diagnosis. VUS variants can be considered as potentially disease-causing variants when they are very close to being classified as LP by bioinformatic analyses and additional evidences, e.g., segregation studies, therapeutic responses to specific treatments, or functional studies, contribute to assigning them a pathogenetic role [14].

To detect exon-level copy number variations (CNVs), the relative quantification of genomic DNA sequences was carried out through an in-house procedure that calculated the ratio between the read counts observed for each exon of the *PHEX* gene (NM_000444.6) and the average read counts of all the exons of a reference autosomal gene (*AGRN*) in the patients relative to the average ratio of the read counts of each *PHEX* exon normalized to the average read counts of all the exons of the agrin gene (*AGRN*) reported in three normal females. Relative gene dosages were expressed as fold changes (two copies = 0.85 − 1.20 and one copy = 0.35 − 0.65) [15]. The data were analyzed and reported as graphs through Excel charts.

The presence of heterozygous deletions/duplications in *PHEX* was further assessed by multiplex ligation probe amplification (MLPA) with the SALSA MLPA Probemix P223 *PHEX* (MRC Holland, Amsterdam, The Netherlands) for the detection of deletions or duplications in the *PHEX* (RefSeq: NM_000444.6) and *FGF23* (RefSeq: NM_020638.3) genes. MLPA data analysis was carried out by the Coffalyser.Net software (MRC Holland, Amsterdam, The Netherlands).

## 3. Results

### 3.1. Probands’ Clinical Features

Two unrelated female infants were referred to the hospital due to their short statures, bowing legs, and waddling gaits that delayed their walking milestones. In both cases, the parents were normal in their heights and physical structures.

The first patient was a 25-month-old girl with a mild short stature (−1.09 SD) and low weight (+0.36 Z-Score), and her lower limb radiographs showed proximal tibialis enlargement (Figure 1A).

Blood tests showed normal kidney function and normal serum calcium levels, though serum phosphate and 25(OH)2D were low and ALP was elevated. Moreover, the low tubular maximum reabsorption rate of phosphate to the glomerular filtration rate (TmP/GFR; normal range 3.25–5.51 mg/dL) indicated renal phosphate wasting (Table 1).

The second patient was a 2-year-old girl. At physical examination, her height (−2.21 SD) and weight (+0.92 Z-Score) were under the median scores (Table 1). Laboratory tests revealed normal kidney function and normal calcium and 25(OH)2D serum levels. Her serum phosphate was low and her ALP was elevated, though her PTH levels were normal. The TmP/GFR level was consistent with renal phosphate wasting (Table 1). X-rays revealed widening of the proximal tibia (not shown). Due to the clinical findings and the unremarkable family history, in this case, a diagnosis of sporadic HR was formulated, and treatment with phosphate supplementation (0.5–1.5 g/day) was started.

Both patients had suboptimal response to traditional therapy, and this was associated with undesired effects such as nausea, vomiting, and, often, diarrhea, abdominal pain, and bloating due to the high doses of oral phosphates.

Subsequent laboratory tests revealed, in both children, elevated serum levels of intact FGF23 (Table 1). Therefore, phosphorus administration was discontinued to start treatment with burosumab (0.8 mg/kg every 15 days), the FGF23-neutralizing monoclonal antibody [16]. After 1 year of therapy, several beneficial effects on biochemical (Table 2), physical, and clinical parameters were observed in both the children.

In addition to increased phosphorus levels and strong decreases in ALP serum levels, both children reported ameliorated muscle pain and mobility, which resulted in decreased waddling gaits and associated levels of fatigue. The linear growth, the radiographic appearance of rickets, the leg deformities, and the bone density levels were also improved (Figure 1B).

### 3.2. Molecular Analysis

Given the difficulties in differentiating the multiple possible genetic causes of HR, to obtain definitive diagnosis, molecular analysis was carried out. In particular, panel-based NGS analysis targeting exons and flanking exonic sequences of approximately 4800 genes associated with specific clinical phenotypes, aka clinical exome, was performed and interrogated for SNVs in the 20 genes currently associated with HR, as reported in the medical literature, as follows: *ALPL*, *CLCN5*, *CTNS*, *CYP2R1*, *CYP27B1*, *CYP3A4*, *DMP1*, *ENPP1*, *FAH*, *FAM20C*, *FGF23*, *FGFR1*, *HNRNPC*, *KLOTHO*, *GNAS*, *PHEX*, *RAS*, *SLC34A1*, *SLC34A3*, and *VDR* [12]. Genetic data were also interrogated to search for variants in genes associated with conditions that include the phenotype HP:0004912, e.g., “hypophosphataemic rickets”, according to the Human Phenotype Ontology (HPO) classification.

Despite that coverage of the exons and flanking splice sites of the targeted HR genes with a minimum read depth of 20x was 100%, the analysis of genomic data did not detect any P/LP or VUS variants. However, based on the clinical and biochemical findings and on the excellent responses to burosumab therapy, we considered the possibility that our sporadic HR patients could be affected by XLH, the most common form of HR, as consequence of heterozygous CNVs in the *PHEX* gene. Therefore, we performed a quantitative analysis of the read counts relative to each exon of this gene in the patients compared to three normal females [15]. Surprisingly, for both children, we obtained relative ratios consistent with the presence of large heterozygous intragenic deletions. In particular, the deletion found in patient 1 removed the *PHEX* exons 21–22 whereas the deletion detected in patient 2 included the *PHEX* exons 15–22 (Figure 2).

MLPA analyses confirmed the deletions in the probands and excluded them in the respective parents, thereby confirming their de novo origin.

## 4. Discussion

Hypophosphataemic rickets (HR) with elevated FGF23 serum level represents a group of rare conditions that can be successfully treated with targeted drug therapies. In fact, the human anti-FGF23 monoclonal antibody burosumab was the first drug approved in 2018 by the Food and Drug Administration and the European Medicines Agency for the treatment of children aged 1–4 years affected by XLH, the X-linked dominant form of HR [2,17,18].

Unlike the conventional therapy for XLH, which aims to counteract the detrimental effects of FGF23 excess with multiple daily doses of oral phosphate supplements and active vitamin D analogues, burosumab directly addresses the molecular mechanism of the condition simply by targeting the deleterious growth factor. Both the treatments are usually initiated at the time of diagnosis and continued at least until growth completion [3]. However, significantly greater improvements in radiological rickets healing and phosphate, ALP, and calcitriol concentrations, as well as TmP/GFR, growth, and functional outcomes, have been observed in children promptly treated with burosumab compared to those treated with the conventional therapy [19,20].

The two children herein presented had clinical diagnoses of sporadic FGF23-related HR and were, therefore, promptly and successfully treated with burosumab, bringing significant clinical improvements from the first months of treatment. However, since HR is a genetically heterogeneous disorder that may be associated with various Mendelian transmission modes, in the absence of a positive family history, molecular testing for the confirmation of clinical suspicion, a differential diagnosis, carrier detection, genetic counseling, and, if appropriate, admission to long-lasting burosumab therapy is strongly recommended [2,3,7].

The use of NGS technology is a reliable and sensitive approach for genetic testing for patients affected by genetically heterogeneous diseases, such as HR. Through the simultaneous analysis of thousands of disease-associated genes, NGS can discover a causative gene and the sequence variants associated with a specific phenotype [21,22]. NGS is highly sensitive in detecting small nucleotide variants (SNVs) in the target regions of analyzed genes. Healthcare professionals are becoming more aware that high throughput molecular screening methodologies are extremely useful, especially for the early diagnoses of rare inherited diseases with approved therapeutic options [23,24].

Therefore, we applied a multi-gene NGS approach for identifying the molecular cause of the disease in our HR patients; however, despite that the target regions of the genes of interest had an excellent coverage (100%, 20x), we did not identify any SNVs that could explain the patients’ phenotypes. Nevertheless, the clinical and laboratory findings were consistent with the possibility that our patients had XLH, which is associated with pathogenic hemizygous/heterozygous null variants in the *PHEX* gene [8,25]. Indeed, the hallmarks of our HR children were the elevated levels of FGF23, which could reasonably have been due to inactivated *PHEX* gene, which is the most common cause of heritable HR [5,8].

The *PHEX* gene spans a genomic region of approximately 210 kb on chromosome Xq 22.11. It contains 22 exons and encodes the phosphate-regulating neutral endopeptidase PHEX, a glycoprotein of 749 amino acids (NP_000435.3) that belongs to the type II integral membrane zinc-dependent endopeptidase family. The PHEX protein has a large extracellular C-terminal domain, which contains the active sites and glycosylation sites, a transmembrane domain, and an N-terminal cytoplasmic tail [26,27]. It is predominantly expressed in osteoblasts, osteocytes, cartilage, and odontoblasts, and in these cells, PHEX deficiencies impair the cellular trafficking, endopeptidase activity, and FGF23 signaling that, in turn, reduce renal phosphate reabsorption, resulting in abnormal bone mineralization and hypophosphatemia [7]. The regulatory mechanism between PHEX and FGF23 remains unclear, but a recent study demonstrated that PHEX is a direct transcriptional inhibitor of the *FGF23* gene [28].

Currently, more than 900 unique pathogenic or likely pathogenic variants have been reported in *PHEX* (NM_000444.6), and more than 85% of them are SNVs, including nonsense, missense, splicing, and frameshift variants (LOVD; https://databases.lovd.nl/shared/variants/PHEX accessed on 30 June 2023; Human gene mutation database, HGMD^®^, accessed on 30 June 2023), which are distributed throughout the whole gene with no hot spots. Approximately 10–15% of the reported mutations are CNVs that are mainly represented by large intragenic deletions [8,25].

Chromosomal deletions of variable sizes that remove single exons or eliminate a whole gene and, sometimes, various contiguous genes are frequent causes of numerous X-linked genetic diseases [29,30]. The relatively high incidence of deletions indicates that chromosome X is predisposed to genomic rearrangements; indeed, X-linked diseases often appear sporadically in a family as consequence of de novo mutational events, including large deletions [29].

Large deletions can be easily detected in hemizygous males by both PCR- and NGS-based DNA analysis; in contrast, heterozygous deletions in carrier females can escape detection because they are masked by the intact allele [13].

To address the possibility that our probands had CNVs, we performed a quantitative analysis of the read counts relative to each exon of *PHEX*. Surprisingly we found that both patients were heterozygous for large intragenic deletions removing, respectively, exons 21–22 and 15–22 of this gene, as was also confirmed by the MLPA analyses. In both cases, the deletions resulted in null alleles that lacked genomic sequences encoding the PHEX extracellular C-terminal domain [27].

Despite that burosumab has been approved for children, adolescents, and adults with XLH, it has shown variable effectiveness [31,32]. Real-word data may provide further insight regarding the potential for early intervention, the adverse events, and the long-term impacts on skeletal development and quality of life in XLH children treated with this new drug [33,34]. Consequently, since XLH remains a condition that severely impacts the quality of life of an affected patient, for families in which a pathogenic *PHEX* variant has been detected, a prenatal molecular diagnosis remains one of the possible prevention options [27,35].

## 5. Conclusions

Our overall data emphasized the importance of considering the clinical and metabolic manifestations of HR when preliminary NGS genetic analysis does not contribute to the diagnosis. Indeed, based on the key clinical features and medical histories of our patients, we expanded the NGS data analyses and, lastly, provided molecular diagnoses by identifying heterozygous large deletions in the *PHEX* gene, a type of genomic variation that is often missed by NGS-based methodologies. Therefore, since the CNVs in *PHEX* genes appear to account for a significant proportion of XLH cases, we recommend evaluating their presence, especially in SNV-negative HR females. A conclusive diagnosis of XLH ensures long-lasting access to burosumab therapy, which was shown to be highly effective in our patients, and it was essential for genetic counseling on the reproductive options in the parents and in the other family members. Lastly, the identification of the molecular alterations associated with XLH contributes to increasing our knowledge about the correlation between the *PHEX* genotypes and disease severity and/or therapy response, which are not yet fully understood [31,32,33,34].

## Figures and Tables

**Figure 1 diagnostics-13-02552-f001:**
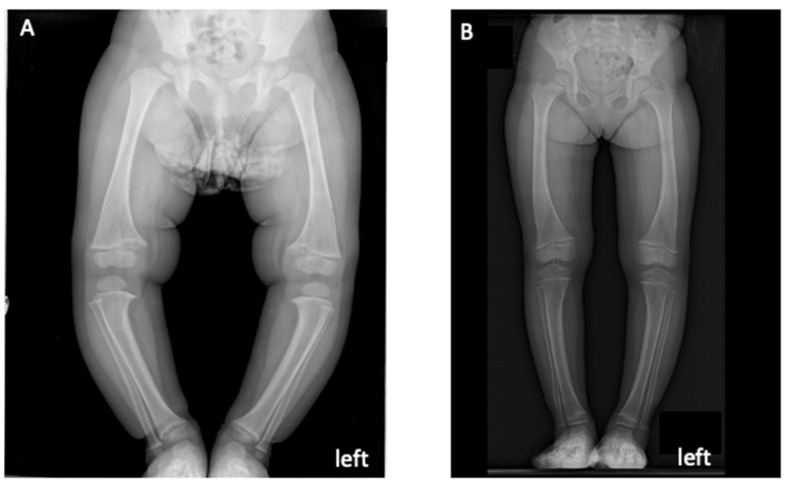
X-ray images showing the lower limb features of patient 1 (**A**) before and (**B**) 1 year after burosomab therapy.

**Figure 2 diagnostics-13-02552-f002:**
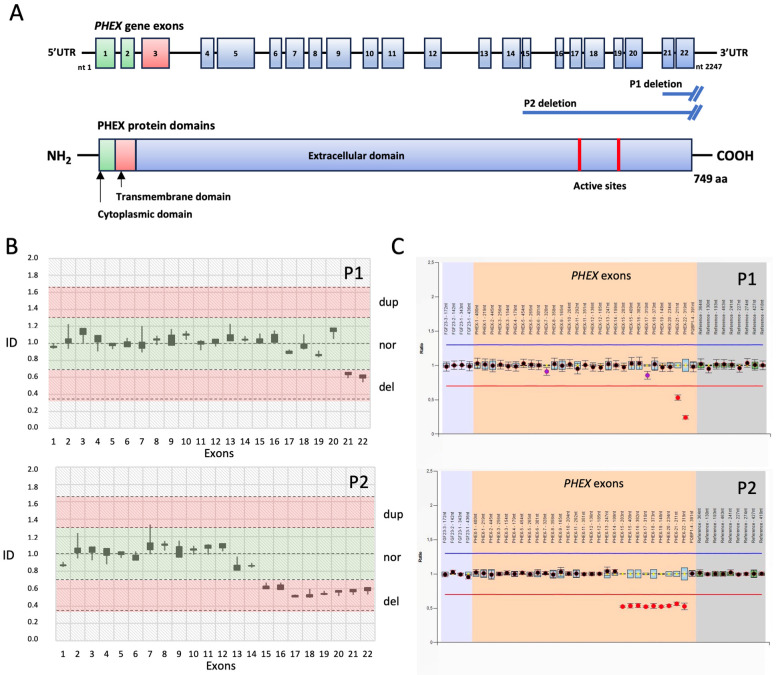
*PHEX* gene structure, protein domains, and the deleted coding sequences. (**A**) Schematic representation of the 22 *PHEX* gene exons and the corresponding protein domains with the approximate positions of the deletions found in the patients (P1 and P2) that removed exons 21–22 and 15–22, respectively, of the gene (downstream regions untested). UTR, untranslated region; nt, nucleotide. (**B**) Boxplot chart of the gene dosage results obtained in patient 1 (P1) and patient 2 (P2) by evaluating the read depths for each of the 22 exons of the *PHEX* gene normalized to the median read depths of all the exons of an autosomal gene (*AGRN*) (see Materials and Methods), which are represented as diagnostic indexes (ID) with respect to three normal controls. The red boxes delimit ID values indicative of heterozygous duplications (dup) or deletions (del) of the analyzed exons. The ID values included within the green box indicate normal copy numbers. The bars represent the SDs of the IDs from the patients normalized to three normal control females. (**C**) Boxplot chart of the MLPA assay results showing the genomic deletions of the *PHEX* gene (red circles in the orange area of the graph) in the two patients.

**Table 1 diagnostics-13-02552-t001:** Patients’ main biochemical and physical findings at presentation.

	Height (cm)	Weight (kg)	eGFR (mL/min/1.73 m^2^)	25(OH)_2_D (ng/mL)	ALP (IU/L)	P (mg/dL)	Ca (mg/dL)	PTH (pg/mL)	FGF23 (ng/L)	TRP (%)	TmP/GFR (mg/dL)
Patient 1	82 ↓	12.7 ↓	113 =	15 ↓	642 ↑	2.2 ↓	8.8 =	18 =	79.5 ↑	77 ↓	1.86 ↓
Patient 2	79 ↓	13.5 ↓	109 =	30 =	808 ↑	2 ↓	9 =	16 =	90 ↑	75 ↓	1.97 ↓

TRP, tubular reabsorption of phosphorus; ↓, lower than the normal range; =, within the normal range; ↑, higher than the normal range (see Materials and Methods).

**Table 2 diagnostics-13-02552-t002:** Patients’ biochemical findings after 1 year of therapy.

	eGFR (mL/min/1.73 m^2^)	25(OH)_2_D (ng/mL)	ALP (IU/L)	P (mg/dL)	Ca (mg/dL)	PTH (pg/mL)	TRP (%)	TmP/GFR (mg/dL)
Patient 1	111	75	259	3.7	9.5	22	85	3.30
Patient 2	194	65	289	3.7	9.1	23	87	3.25

TRP, tubular reabsorption of phosphorus. All parameters were within the respective normal ranges (see Materials and Methods).

## Data Availability

The data that support the findings of this study are available within the article.

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
