# Peer review of "De Novo Large Deletions in the PHEX Gene Caused X-Linked Hypophosphataemic Rickets in Two Italian Female Infants Successfully Treated with Burosumab"

_diagnostics, 2023, doi:10.3390/diagnostics13152552_

Round 1
Reviewer 1 Report
Pecoraro et al reported on the identification of two de novo large genomic deletions of the PHEX gene in two XLH cases.
The paper is interesting and well written.
Minor revisions
Abstract: line 25. “Diagnosis of was based…”: please delete “of”.
Line 68: “(OMIM: #605380)”. The code should be deleted or, better, added when the FGF23 gene is indicated for the first time into the text (line 42).
Table 1. Please indicate the normal ranges for all the biochemical parameters.
Table 2. Values are the same as in the table 1. Please correct.
Figure 2. It would be of help adding also a picture of the boxplot chart of the MLPA assay showing the genomic deletions, in the same figure along with the gene dosage charts.
Author Response
We are very grateful to the reviewer for the kind and precious comments. The comments/ suggestions have been carefully considered and the manuscript revised accordingly.
- Abstract: line 25. “Diagnosis of was based…”: please delete “of”.
R1. We have removed the word, which was a printing error. Thank you for noticing.
- Line 68: “(OMIM: #605380)”. The code should be deleted or, better, added when the FGF23 gene is indicated for the first time into the text (line 42).
R2. Thanks for your remark. Although the FGF23 gene is mentioned for the first time right in the old line 68 (line 77 of the revised manuscript), to make the text uniform, we have listed the main HR-associated genes with the respective OMIM codes in lines 92-103 of the new manuscript.
- Table 1. Please indicate the normal ranges for all the biochemical parameters.
R3. Thank you for your very useful suggestion. We have reported in the Materials and Methods section (lines 134-140) the normal ranges of the biochemical parameters shown in the Tables. Furthermore, to make easy the reading of the data reported in the Tables, we have added appropriate legends.
- Table 2. Values are the same as in the table 1. Please correct.
R4. Thank you very much for noticing this relevant mistake. We have reported the right parameters in the Table 2.
- Figure 2. It would be of help adding also a picture of the boxplot chart of the MLPA assay showing the genomic deletions, in the same figure along with the gene dosage charts.
R5. Thank you for giving us this important suggestion. We have modified the Figure 2 accordingly.
===========================================
Reviewer 2 Report
Dear Authors,
It was a pleasure to review your article, "De novo large deletions in the PHEX gene cause X-linked hypophosphataemic rickets in two Italian female infants successfully treated with burosumab". The article describes two rare cases of hypophosphatemic rickets caused by a rare type of genetic variant in the PHEX gene and the implications for treatment.
I thought there may be several aspects that can improve the content of the article and increase reader's interest.
1) The conclusion stated in the introduction seems rather general and does not show the learning points from these two cases. I thought it may be more useful to mention the need for phenotype-oriented genetic testing, guided by the known types of variants, including the rare types. While collaboration between different professionals is always useful and desired, the diagnostic approach is more interesting for potential readers.
2) In the 'Methods' section, you state: "In agreement, variants can be classified as pathogenic (P), likely pathogenic (LP), variant(s) of uncertain significance (VUS), likely benign (LB), or benign (B). Only P/LP or exonic VUS variants had to be considered for molecular diagnosis.". In fact, only P/LP variants can be considered to confirm the diagnosis. Variants of unknown significance (VUS) are not usually reported, unless they are very close to being classified as LP (so called 'hot VUSs') and additional evidence can help with this. Examples can be additional seggregation studies, therapeutic reponse to specific treatment or functional studies. However, this is always a balanced and multidisciplinary decision and it is rather an exception.
3) R268: "clinical and laboratory findings were consistent with the possibility that our patients, despite unremarkable family history, had 268 XLH". A negative family history does not exclude a genetic diagnosis and cannot represent an argument against a clear clinical diagnosis. By adding "despite unremarkable family history" readers less familiar with genetic terms can get the false impression that this is a relevant argument against the diagnosis so I suggest to remove this part of the sentence.
4) I think it would be useful to add a figure showing the PHEX gene structure, coding regions and the corresponding domains versus the two deletions. This could be included in Figure 2.
Dear Authors,
It was a pleasure to review your article describing a rare type of genetic variant in PHEX gene and patients' reponse to treatment.
The article is generally well written. However, I noticed a few sentences that may need some corrections:
R25: “Diagnosis of was based on clinical and laboratory findings, and physical examination". It seems that one or more words are missing and the sentence would benefit from minor corrections.
R28: “While conventional analysis of NGS data did not reveal pathogenic small nucleotide variants (SNVs) the HR-related genes ...”. This sentence also seems to lack one or more words.
R306: “... further information needs about the potential for early intervention”. I think this part of the sentence needs minor corrections.
R165: "The first patient was a 25-month-old girl, with low deficit in height". Here "mild short stature" wold be more appropriate than "low deficit in height".
R323: instead of "determining proceative risk", I think "informing reproductive options" would be more appropriate.
Author Response
We greatly appreciate your review and therefore thank you very much for your kind and precious comments.
Point-by-point response to the comments:
- The conclusion stated in the introduction seems rather general and does not show the learning points from these two cases. I thought it may be more useful to mention the need for phenotype-oriented genetic testing, guided by the known types of variants, including the rare types. While collaboration between different professionals is always useful and desired, the diagnostic approach is more interesting for potential readers.
R1. Thank you very much for expressing this interesting point of view, which we wholeheartedly share. Therefore, we have changed the sentence that now reads “In conclusion, the execution of a phenotype-oriented genetic test, guided by the known types of variants, including the rarest ones, was crucial to reach the definitive diagnosis and ensure our patients a long-term therapy administration.”, just making your words ours (lines 34-36).
- In the 'Methods' section, you state: "In agreement, variants can be classified as pathogenic (P), likely pathogenic (LP), variant(s) of uncertain significance (VUS), likely benign (LB), or benign (B). Only P/LP or exonic VUS variants had to be considered for molecular diagnosis.". In fact, only P/LP variants can be considered to confirm the diagnosis. Variants of unknown significance (VUS) are not usually reported, unless they are very close to being classified as LP (so called 'hot VUSs') and additional evidence can help with this. Examples can be additional seggregation studies, therapeutic reponse to specific treatment or functional studies. However, this is always a balanced and multidisciplinary decision and it is rather an exception.
R2. We fully agree with this statement, and we thank you for this clarification, which therefore has been included in the new manuscript (lines 165-169).
- R268: "clinical and laboratory findings were consistent with the possibility that our patients, despite unremarkable family history, had 268 XLH". A negative family history does not exclude a genetic diagnosis and cannot represent an argument against a clear clinical diagnosis. By adding "despite unremarkable family history"readers less familiar with genetic terms can get the false impression that this is a relevant argument against the diagnosis so I suggest to remove this part of the sentence.
R3. Thank you for your suggestion. We have removed the sentence.
- I think it would be useful to add a figure showing the PHEX gene structure, coding regions and the corresponding domains versus the two deletions. This could be included in Figure 2.
R4. Thank you for giving us this useful suggestion. We have modified the Figure 2 accordingly.
Comments on the Quality of English Language
R25: “Diagnosis of was based on clinical and laboratory findings, and physical examination". It seems that one or more words are missing and the sentence would benefit from minor corrections.
R. We have removed the word “of”, which was a printing error. Thank you for noticing.
R28: “While conventional analysis of NGS data did not reveal pathogenic small nucleotide variants (SNVs) the HR-related genes ...”. This sentence also seems to lack one or more words.
R. We have added the lacking words. Thank you for noticing.
R306: “... further information needs about the potential for early intervention”. I think this part of the sentence needs minor corrections.
R. Thank you for observation. We have changed the sentence and added a pertinent reference (n. 34 in the revised manuscript).
R165: "The first patient was a 25-month-old girl, with low deficit in height". Here "mild short stature" wold be more appropriate than "low deficit in height".
R. We have accepted your kind suggestion and changed the sentence accordingly.
R323: instead of "determining proceative risk", I think "informing reproductive options" would be more appropriate.
R. We have accepted this further useful suggestion and changed the sentence accordingly.